# Ctrl-Room: Controllable Text-to-3D Room Meshes Generation with Layout Constraints

## Abstract

Text-driven 3D indoor scene generation could be useful for gaming, film industry, and AR/VR applications. However, existing methods cannot faithfully capture the room layout, nor do they allow flexible editing of individual objects in the room. To address these problems, we present Ctrl-Room, which is able to generate convincing 3D rooms with designer-style layouts and high-fidelity textures from just a text prompt. Moreover, Ctrl-Room enables versatile interactive editing operations such as resizing or moving individual furniture items. Our key insight is to separate the modeling of layouts and appearance. Our proposed method consists of two stages, a 'Layout Generation Stage' and an 'Appearance Generation Stage'. The 'Layout Generation Stage' trains a text-conditional diffusion model to learn the layout distribution with our holistic scene code parameterization. Next, the 'Appearance Generation Stage' employs a fine-tuned ControlNet to produce a vivid panoramic image of the room guided by the 3D scene layout and text prompt. In this way, we achieve a high-quality 3D room with convincing layouts and lively textures. Benefiting from the scene code parameterization, we can easily edit the generated room model through our mask-guided editing module, without expensive editing-specific training. Extensive experiments on the Structured3D dataset demonstrate that our method outperforms existing methods in producing more reasonable, view-consistent, and editable 3D rooms from natural language prompts.

## 1 Introduction

High-quality textured 3D models are important for a broad range of applications, from interior design and games to simulators for embodied AI. Indoor scenes are of particular interest among all 3D content. Typically, 3D indoor scenes are manually designed by professional artists, which is time-consuming and expensive. While recent advancements in generative models (Poole et al., 2022; Chen et al., 2023; Lin et al., 2023; Seo et al., 2023) have simplified the creation of 3D models from textual descriptions, extending this capability to text-driven 3D indoor scene generation remains a challenge because indoor scenes exhibit strong semantic layout constraints, e.g., neighboring walls are perpendicular and the TV set often faces a sofa, which are more complicated than objects.

Existing text-driven 3D indoor scene generation approaches, such as Text2Room (Höllein et al., 2023) and Text2NeRF (Zhang et al., 2023), are designed with an incremental framework. They create 3D indoor scenes by incrementally generating different viewpoints frame-by-frame and reconstructing the 3D mesh of the room from these sub-view images. However, their incremental approaches often fail to model the global layout of the room, resulting in unconvincing results that lack semantic plausibility. As shown in the left of Fig. 1 (a), the result of Tex2Room exhibits repeating objects, e.g. several cabinets in a living room, and does not follow the furniture layout patterns. We refer to this problem as the *'Penrose Triangle problem'* in our paper, which has plausible 3D structures everywhere locally but lacks global consistency. Moreover, previous methods fail to enable user interactive manipulation as their resulting 3D geometry and texture are uneditable.

Indoor scenes might also be represented by a panorama image. Several works (Lin et al., 2019; 2021; Shum et al., 2023; Tang et al., 2023c) have been proposed to generate such a panorama from a text prompt. We might further recover the depth map of these images to build a textured 3D room model. However, these works also cannot guarantee correct room layouts. As shown on the right of Fig. 1 (a), a bedroom generated by MVDiffusion (Tang et al., 2023c) contains multiple beds,

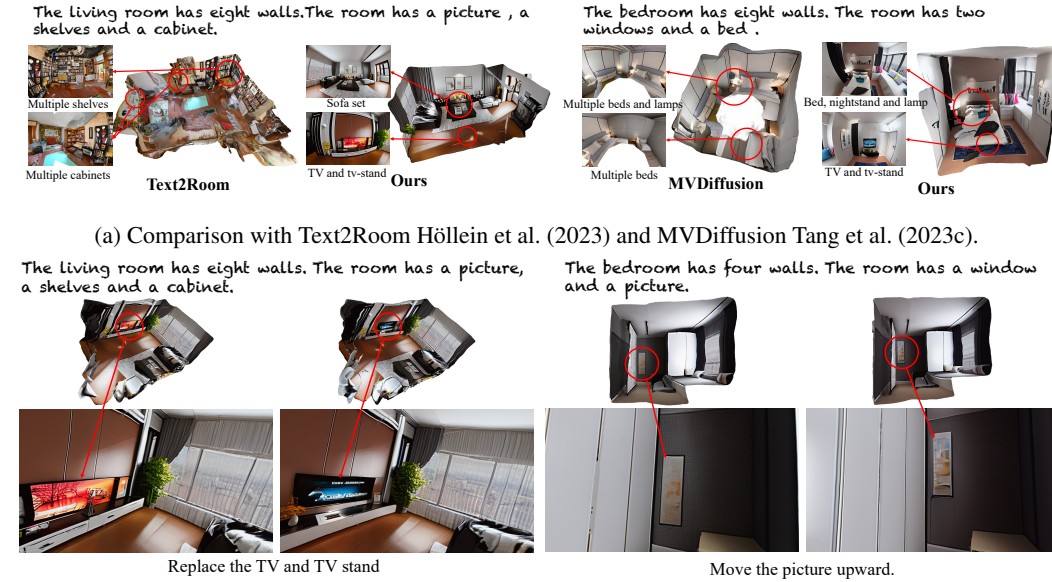

(a) Comparison with Text2Room Höllein et al. (2023) and MVDiffusion Tang et al. (2023c).

(b) Flexible editing by instruction or mouse clicks.

Figure 1: We present *Ctrl-Room* to achieve fine-grained textured 3D indoor room generation and editing. (a): compared with the Text2Room (Höllein et al., 2023) and MVDiffusion(Tang et al., 2023c), Ctrl-Room can generate rooms with more plausible 3D structures. (b): Ctrl-Room supports flexible editing. Users can replace furniture items or change their positions easily.

which also violates the room layout prior. Furthermore, these methods also cannot easily control the individual room objects.

To address these shortcomings, we propose a novel two-stage method to generate a high-fidelity and editable 3D room. The key insight is to separate the generation of 3D geometric layouts from that of the visual appearance, which allows us to better capture the room layout and achieve flexible editing at the same time. In the first stage, from text input, our method creates plausible room layouts with various furniture types and positions. Unlike previous scene synthesis methods (Tang et al., 2023a; Paschalidou et al., 2021) that only focus on the furniture arrangement, our approach further considers walls with doors and windows, which play an essential role in the layout. To achieve this goal, we parameterize the room by a holistic scene code, which represents a room as a set of objects. Each object is represented by a vector capturing its position, size, semantic class, and orientation. Based on our compact parameterization, we design a diffusion model to learn the 3D room layout distribution from the Structured3D dataset (Zheng et al., 2020).

Once the indoor scene layout is fixed, in the second stage, our method generates the room appearance with the guidance of the 3D room layout. We model the appearance of the room as a panoramic image, which is generated by a text-to-image latent diffusion model. Unlike previous text-to-panorama works (Tang et al., 2023c; Chen et al., 2022), our method explicitly enforces room layout constraints and guarantees plausible 3D room structures and furniture arrangement. To achieve this goal, we convert the 3D layout synthesized in the first stage into a semantic segmentation map and feed it to a fine-tuned ControlNet (Zhang & Agrawala, 2023) model to create the panorama image.

Most importantly, benefiting from the separation of layout and appearance, our method enables flexible editing on the generated 3D room. The user can replace or modify the size and position of furniture items, e.g. replacing the TV and TV stand or moving up the picture as in Fig. 1 (b), by instructions or mouse clicks. Our method can quickly update the room according to the edited room layout through our mask-guided editing module without expensive editing-specific training. The updated room appearance maintains consistency with the original version while satisfying the user's edits. To our knowledge, it's the first work that achieves 3D indoor scene editing through a 2D diffusion model.

The main contributions of this paper are summarized as:

- To address the Penrose Triangle Problem, We design a two-stage method for 3D room generation from text input, which separates the geometric layout generation and appearance generation. In this way, our method can better capture the room layout constraints in real-world data and produce a vivid and rich appearance at the same time.

- Our separation of geometric layout and visual appearance allows us to have flexible control and editing over the generated 3D room model. Users can adjust the size, semantic class, and position of furniture items easily.

- We introduce a novel method to generate and edit panoramic images, which achieves high-quality results with loop consistency through a pre-trained latent image diffusion model without expensive editing-specific training.

## 2 RELATED WORK

### 2.1 TEXT-BASED 3D OBJECT GENERATION

Early methods employ 3D datasets to train generation models. Chen et al. (2019) learn a feature representation from paired text and 3D data and design a GAN network to generate 3D shapes from text. However, 3D datasets are scarce which makes these methods difficult to scale. More recent methods (Nichol et al., 2022; Poole et al., 2022; Lin et al., 2023; Wang et al., 2023a; Chen et al., 2023; Wang et al., 2023b) exploit the powerful 2D text-to-image diffusion models (Rombach et al., 2022; Saharia et al., 2022) for 3D model generation. Typically, these methods generate one or multiple 2D images in an incremental fashion and optimize the 3D model accordingly. Point-E (Nichol et al., 2022) employs a text-to-image diffusion model (Rombach et al., 2022) to generate a single-view image for a point cloud diffusion model to generate a 3D point cloud. DreamFusion (Poole et al., 2022) introduces a loss based on probability density distillation and optimizes a randomly initialized 3D model through gradient descent. Magic3D (Lin et al., 2023) uses a coarse model to represent 3D content and accelerates it using a sparse 3D hash grid structure. Wang et al. (2023a) use Score Jacobian Chaining to aggregate the results of the 2D diffusion models to generate a 3D scene. Fantasia3D (Chen et al., 2023) optimizes a mesh from scratch with DMTet (Shen et al., 2021) and stable diffusion (Rombach et al., 2022). To alleviate over-saturation, over-smoothing, and low-diversity problems, ProlificDreamer (Wang et al., 2023b) models and optimizes the 3D parameters, NeRF (Mildenhall et al., 2021) or mesh, through variational score distillation. However, all these methods focus on text-based 3D object generation. They cannot be directly applied to create 3D rooms that have additional structural layout constraints.

### 2.2 TEXT-BASED 3D ROOM GENERATION

**Room Layout Synthesis** Layout generation has been greatly boosted by transformer-based methods. LayoutTransformer (Gupta et al., 2021) employs self-attention to capture relationships between elements and then accomplish layout completion. ATISS (Paschalidou et al., 2021) proposes an autoregressive transformer to generate proper indoor scenes with only the room type and floor plan as the input. Recently, DiffuScene (Tang et al., 2023a) uses a fully connected scene graph to represent a scene and proposes a diffusion model to generate all objects in the scene. These methods only focus on the relative positions of different furniture items and ignore their appearance. Furthermore, unlike our methods, they do not consider walls, doors, and windows when learning layouts, which play important roles in the arrangement of real furniture.

**Panoramic Image Generation** Another line of works (Lin et al., 2019; 2021; Shum et al., 2023) represents an indoor scene by a panorama image without modeling 3D shapes. These methods enjoy the benefits of abundant image training data and produce vivid results. COCO-GAN (Lin et al., 2019) produces a set of patches and assemble them into a panoramic image. InfinityGAN (Lin et al., 2021) uses the information of two patches to generate the parts between them, and finally obtains a panoramic image. Shum et al. (2023) proposes a 360-aware layout generator to produce furniture arrangements and uses this layout to synthesize a panoramic image based on the input scene background. MVDiffusion (Tang et al., 2023c) simultaneously generates multi-view perspective images and proposes a correspondence-aware attention block to maintain multi-view consistency, and then transfers these images to a panorama. These methods might suffer from incorrect room

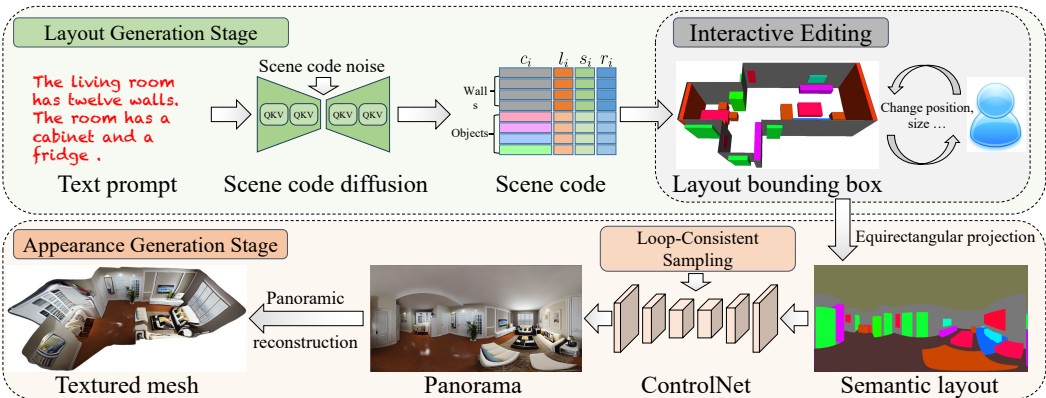

Figure 2: Overview of our method. In the Layout Generation Stage, we synthesize a scene code from the text input and convert it to a 3D bounding box representation to facilitate editing. In the Appearance Generation Stage, we project the bounding boxes into a semantic segmentation map to guide the panorama synthesis. The panorama is then reconstructed into a textured 3D mesh model.

layout since they do not enforce layout constraints. Furthermore, the results of these methods cannot be easily edited, e.g. resizing or moving some furniture items around, because they do not maintain an object-level representation.

**3D Room Generation** There are also 3D room generation methods. GAUDI (Bautista et al., 2022) generates immersive 3D indoor scenes rendered from a moving camera. It disentangles the 3D representation and camera poses to ensure the consistency of the scene during camera movement. CC3D (Bahmani et al., 2023) trains a 3D-aware GAN for multi-object scenes conditioned on a semantic layout image, while it still relies on multi-view supervision during training. Room-Dreamer (Song et al., 2023) employs 2D diffusion models to stylize and improve a given mesh. Text2room (Höllein et al., 2023) incrementally synthesizes nearby images with a 2D diffusion model and recovers its depth maps to stitch these images into a 3D room model. Text2Room is the closest to our work, but it cannot handle the geometric and textural consistency among multi-posed images, resulting in the *'Penrose Triangle problem'*. In our method, we take both geometry and appearance into consideration and create a more geometrically plausible 3D room.

## 2.3 DIFFUSION-BASED IMAGE EDITING

Diffusion models have been employed in image editing and produced inspiring results. Prompt-to-prompt (Hertz et al., 2022) requires only text input and modifies the attention map within pre-trained diffusion models to achieve image editing capabilities. Imagic (Kawar et al., 2023) generates an edited image that aligns with the given input text through semantic optimization of text embedding within the image. But these editing methods all modify images at a high level (through text). Inspired by the idea that latent code can determine the spatial distribution of generated images (Mao et al., 2023), DragDiffusion (Shi et al., 2023) enables users to perform pixel-level spatial control on images by allowing pixel dragging. In contrast, DragonDiffusion (Mou et al., 2023) requires no training or further fine-tuning and all consistency information is from the image. Only relying on the information inside an image limits flexible editing. In pursuit of better performance and control, we additionally introduce semantic segmentation maps generated from the room layout to guide the panorama image generation to achieve flexible control.

## 3 METHOD

In order to achieve text-based 3D indoor scene generation and editing, we propose our Ctrl-Room. We first generate the room layout from the input text and then generate the room appearance according to the layout, followed by panoramic reconstruction to generate the final 3D textured mesh. This mechanism enables users to interactively edit the scene layout so as to produce customized 3D mesh results. The overall framework of our method is depicted in Fig. 2, which consists of two stages:

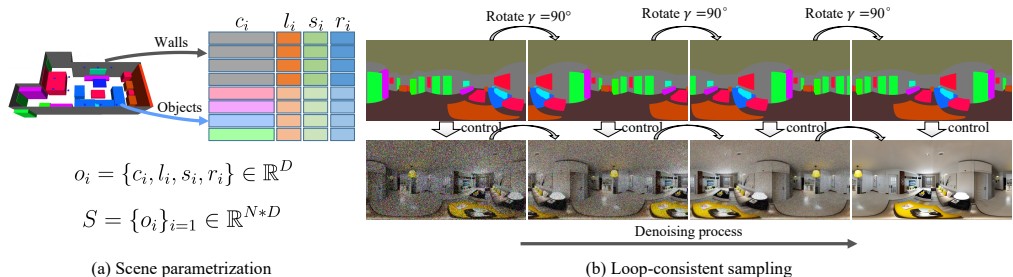

$o_i = \{c_i, l_i, s_i, r_i\} \in \mathbb{R}^D$

$S = \{o_i\}_{i=1} \in \mathbb{R}^{N*D}$

(a) Scene parametrization               (b) Loop-consistent sampling

Figure 3: (a) A 3D scene $S$ is represented by its scene code $x_0 = \{o_i\}_{i=1}^N$, where each wall or furniture item $o_i$ is a row vector storing attributes like class label $c_i$, location $l_i$, size $s_i$, orientation $r_i$. (b) During the denoising process, we rotate both the input semantic layout panorama and the denoised image for $\gamma$ degree at each step. Here we take $\gamma = 90°$ for example.

the Layout Generation Stage and the Appearance Generation Stage. In the Layout Generation Stage, we use a holistic scene code to parametrize the indoor scene and design a diffusion model to learn its distribution. Once the holistic scene code is generated from text, we recover the room as a set of orientated bounding boxes of walls and objects. Note that users can edit these bounding boxes by dragging objects to adjust their semantic types, positions, or scales, enabling the customization of 3D scene results according to the user's preferences. In the Appearance Generation Stage, we obtain an RGB panorama through a pre-trained latent diffusion model to represent the room texture. Specifically, we project the generated layout bounding boxes into a semantic segmentation map representing the layout. We then fine-tune a pre-trained ControlNet (Zhang & Agrawala, 2023) model to generate an RGB panorama from the input layout panorama. To ensure loop consistency, we propose a novel loop-consistent sampling during the inference process. Finally, the textured 3D mesh is obtained by estimating the depth map of the generated panorama. By separating the scene layout and appearance, our framework supports diverse editing operations on the 3D scene.

## 3.1 LAYOUT GENERATION STAGE

**Scene Code Definition.** Different from previous methods (Paschalidou et al., 2021; Tang et al., 2023a), we consider not only furniture but also walls, doors, and windows to define the room layout. We employ a unified encoding of various objects. Specifically, given a 3D scene $\mathcal{S}$ with $m$ walls and $n$ furniture items, we represent the scene layout as a holistic scene code $\mathbf{x_0} = \{\mathbf{o_i}\}_{i=1}^N$, where $N = m + n$. We encode each object $o_j$ as a node with various attributes, i.e., center location $l_i \in \mathbb{R}^3$, size $s_i \in \mathbb{R}^3$, orientation $r_i \in \mathbb{R}$, class label $c_i \in \mathbb{R}^C$. Each node is characterized by the concatenation of these attributes as $\mathbf{o_i} = [c_i, l_i, s_i, r_i]$. As can be seen in Fig. 3 (a), we represent a scene layout as a tensor $\mathbf{x_0} \in \mathbb{R}^{N \times D}$, where $D$ is the attribute dimension of a node. In all the data, we choose the normal direction of the largest wall as the 'main direction'. For other objects, we take the angles between their front directions and the main direction as their rotations. We use the one-hot encoding to represent their semantic types, such as sofa or lamp. For more details on the scene encoding, please refer to the Appendix.

**Scene Code Diffusion.** With the scene code definition, we build a diffusion model to learn its distribution. A scene layout is a point in $\mathbb{R}^{N \times D}$. The forward diffusion process is a discrete-time Markov chain in $\mathbb{R}^{N \times D}$. Given a clean scene code $\mathbf{x}_0$, the diffusion process gradually adds Gaussian noise to $\mathbf{x}_0$, until the resulting distribution is Gaussian, according to a pre-defined, linearly increased noise schedule $\beta_1, ..., \beta_T$:

$$q(\mathbf{x_t}|\mathbf{x_0}) := \mathcal{N}(\mathbf{x_t}; \sqrt{\bar{\alpha}_t}\mathbf{x_0}, (1 - \sqrt{\bar{\alpha}_t})\mathbf{I}) \qquad (1)$$

where $\alpha_t := 1 - \beta_t$ and $\bar{\alpha}_t := \prod_{r=1}^t \alpha_r$ define the noise level and decrease over the timestep $t$. A neural network is trained to reverse that process, by minimizing the denoising objective,

$$\mathcal{L} = \mathbf{E}_{\mathbf{x_0},t,y,\epsilon}\|\epsilon - \epsilon_\theta(x_t, t, y)\|^2, \qquad (2)$$

where $\epsilon_\theta$ is the noise estimator which aims to find the noise $\epsilon$ added into the input $x_0$. Here, $y$ is the text embedding of the input text prompts. The denoising network is a 1D UNet (Ronneberger et al., 2015), with multiple self-attention and cross-attention layers designed for input text prompts. The

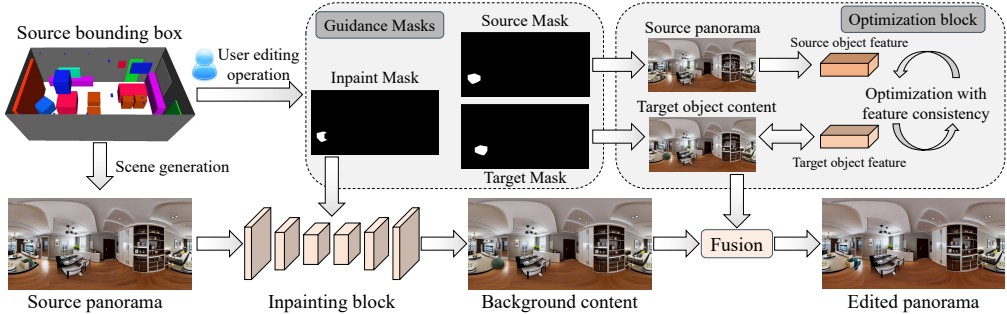

Figure 4: **Mask-guided Editing.** After editing the scene bounding box, we derive guidance masks from the changes in the semantic layout panoramas. We fill in unoccluded regions and optimize the DIFT (Tang et al., 2023b) features to keep the identity of moved objects unchanged.

denoising network $\epsilon_\theta$ takes the scene code $\mathbf{x_t}$, text prompt $y$, and timestep $t$ as input, and denoises them iteratively to get a clean scene code $\hat{\mathbf{x}}_0$. Then we represent $\hat{\mathbf{x}}_0$ as a set of orientated bounding boxes of various semantic types to facilitate interactive editing.

## 3.2 APPEARANCE GENERATION STAGE

Given the layout of an indoor scene, we seek to obtain a proper panorama image to represent its appearance. Instead of incrementally generating multi-view images like (Höllein et al., 2023), we generate the entire panorama at once. We utilize ControlNet (Zhang & Agrawala, 2023) to generate a high-fidelity panorama conditioned by the input 3D scene layout. After getting the scene panorama, we recover the depth map by the method in (Shen et al., 2022) to reconstruct a textured mesh through Possion reconstruction (Kazhdan et al., 2006) and MVS-texture (Waechter et al., 2014).

**Fine-tune ControlNet.** ControlNet is a refined Stable Diffusion (Rombach et al., 2022) model conditioned on an extra 2D input. To condition ControlNet on the scene layout, we convert the bounding box representation into a 2D semantic layout panorama through equirectangular projection. In this way, we get a pair of RGB and semantic layout panoramic images for each scene. However, the pre-trained ControlNet-Segmentation (github, 2023) is designed for perspective images, and cannot be directly applied to panoramas. Thus, we fine-tune it with our pairwise RGB-Semantic layout panoramas on the Structured3D dataset Zheng et al. (2020). As the volume of Structured3D is limited, we apply several techniques during the fine-tuning to augment the training data, including standard left-right flipping, horizontal rotation, and Pano-Stretch (Sun et al., 2019).

**Loop-consistent Sampling.** A panorama should be loop-consistent. In other words, its left and right should be seamlessly connected. Although the panoramic horizontal rotation in data augmentation may improve the model's implicit understanding of the expected loop consistency, it lacks explicit constraints and might still produce inconsistent results. Therefore, we propose an explicit loop-consistent sampling mechanism in the denoising process of the latent diffusion model. As shown in Fig. 3(b), we rotate both the input layout panorama and the denoised image for $\gamma$ degree in the sampling process, which applies explicit constraints for the loop consistency during denoising. A concurrent work (Wu et al., 2023) also uses a similar method for panoramic outpainting. More qualitative results in Appendix Figure.2 verify that our simple loop-consistent sampling method achieves good results without introducing additional learnable parameters.

## 3.3 MASK-GUIDED EDITING

A user can modify the generated 3D room by changing the position, semantic class, and size of object bounding boxes. Our method will update the panorama accordingly to reflect the user's editing in 3D space, which achieves zero-shot editing without expensive re-training.

The editing should achieve two goals, i.e. altering the content according to the user's input, and maintaining appearance consistency for scene objects. We propose a mask-guided image editing as illustrated Fig. 4, where a chair's position is moved. In the following, we will explain our method with this example. We denote the semantic panorama from the edited scene as $P_{\text{edited}}$, then we derive the guidance masks based on its difference from the original one $P_{\text{ori}}$. The source mask

$\mathbf{m}_{\text{src}}$ shows the position of the original chair, and the target mask $\mathbf{m}_{\text{tar}}$ indicates the location of the moved chair, and the inpainting mask $\mathbf{m}_{\text{inpaint}} = \{m | m \in \mathbf{m}_{\text{src}} \text{ and } m \notin \mathbf{m}_{\text{tar}}\}$ is the unoccluded region. Given these guidance masks, our method includes two steps: the inpainting step and the optimization step. We first fill in the inpaint area by feeding the inpaint mask $\mathbf{m}_{\text{inpaint}}$ and edited semantic panorama $P_{\text{edited}}$ to the inpainting step. Then, in our optimization step, we optimize the DIFT (Tang et al., 2023b) feature to maintain the visual consistency of relocated objects.

**Inpainting Step.** Denoting the original image as $\mathbf{x}_0^{\text{ori}}$, we replace pixels outside the inpainting mask $\mathbf{m}_{\text{inpaint}}$ with $\mathbf{x}_t^{\text{ori}}$ during the diffusion process. This simple strategy keeps the outside region unchanged. At each reverse diffusion step, we compute:

$$\mathbf{x}_t^{\text{ori}} \sim \mathcal{N}(\sqrt{\bar{\alpha}_t}\mathbf{x}_0^{\text{ori}}, (1 - \bar{\alpha}_t\mathbf{I})), \tag{3}$$

$$\mathbf{x}_t^{\text{new}} \sim \mathcal{N}(\mu_\theta(x_t, t, y, P_{\text{edited}}), \Sigma_\theta(x_t, t, y, P_{\text{edited}})), \tag{4}$$

$$\hat{\mathbf{x}}_{t-1}^{\text{new}} = \mathbf{m}_{\text{inpaint}} \odot \mathbf{x}_t^{\text{new}} + (1 - \mathbf{m}_{\text{inpaint}}) \odot \mathbf{x}_t^{\text{ori}}, \tag{5}$$

where $\mathbf{x}_t^{\text{ori}}$ is obtained through propagating $\mathbf{x}_0^{\text{ori}}$ in diffusion process, and $\mathbf{x}_t^{\text{new}}$ is sampled from the fine-tuned ControlNet model, which takes the edited semantic layout panorama $P_{\text{edited}}$ and text prompt $y$ as input. As the propagated $\mathbf{x}_t^{\text{ori}}$ is unaware of the new content $\mathbf{x}_t^{\text{new}}$, this may result in distracting boundaries of the inpainted area. To better blend the new content $\mathbf{x}_t^{\text{new}}$ and its surrounding background $\mathbf{x}_t^{\text{ori}}$ in the inpainted area, we update the computation of $\hat{\mathbf{x}}_{t-1}^{\text{new}}$ to,

$$\hat{\mathbf{x}}_{t-1}^{\text{new}} = \mathbf{m}_{\text{inpaint}} \odot \mathbf{x}_t^{\text{new}} + (1 - \mathbf{m}_{\text{inpaint}}) \odot (\mathbf{x}_t^{\text{ori}} \cdot \lambda_{\text{ori}} + \mathbf{x}_{t+1}^{\text{new}} \cdot \lambda_{\text{new}}), \tag{6}$$

where $\lambda_{\text{ori}}$ and $\lambda_{\text{new}}$ are hyper-parameters to adjust the weight for fusing the inpainted area and unchanged area. The final result of inpainting is $\hat{\mathbf{x}}_0^{\text{new}}$.

**Optimization Step.** When the user moves the position of a furniture item, we need to keep its appearance unchanged before and after the movement. The recent work, DIFT (Tang et al., 2023b), finds the learned features from the diffusion network allow for strong semantic correspondence. Thus, we maintain the consistency between the original and moved furniture by requiring their latent features to be consistent. In particular, we extract latent features $F_t^l$ of the layer $l$ in the denoising U-Net network, at timestep $t$. Then we construct a loss function using the latent features from source area $\mathbf{m}_{\text{src}}$ in source panorama $\mathbf{x}_0^{\text{ori}}$ and target area $\mathbf{m}_{\text{tar}}$ in inpainted panorama $\hat{\mathbf{x}}_0^{\text{new}}$.

For conciseness, we denote the target image $\hat{\mathbf{x}}_0^{\text{edit}}$ initialized by $\hat{\mathbf{x}}_0^{\text{new}}$. We first propagate the original image $\mathbf{x}_0^{\text{ori}}$ and $\hat{\mathbf{x}}_0^{\text{edit}}$ to get $\mathbf{x}_t^{\text{ori}}$ and $\hat{\mathbf{x}}_t^{\text{edit}}$ at timestep $t$ by diffusion process, respectively. At each iteration, we use the same ControlNet model to denoise both $\mathbf{x}_t^{\text{ori}}$ and $\hat{\mathbf{x}}_t^{\text{edit}}$ and extract the latent features of them, denoted as $F_t^{\text{ori}}$ and $F_t^{\text{edit}}$, respectively. Based on the strong correspondence between the features, the source mask area $\mathbf{m}_{\text{src}}$ and the target area $\mathbf{m}_{\text{tar}}$ in $F_t^{\text{ori}}$ and $F_t^{\text{edit}}$ need to have high similarity. Here, we utilize the cosine embedding loss to measure the similarity, and define the optimization loss function as follows:

$$\mathcal{L}_{\text{obj}} = -cos(\text{sg}(F_t^{ori} \odot \mathbf{m}_{\text{src}}), F_t^{edit} \odot \mathbf{m}_{\text{tar}}). \tag{7}$$

Here, sg is the stop gradient operator, the gradient will not be back-propagated for the term $\text{sg}(F_t^{ori} \odot \mathbf{m}_{\text{src}})$. Then we minimize the loss iteratively. At each iteration, $\hat{\mathbf{x}}_t^{\text{edit}}$ is updated by taking one gradient descent step with a learning rate $\eta$ to minimize the loss $\mathcal{L}_{\text{obj}}$ as,

$$\hat{\mathbf{x}}_t^{k+1} = \hat{\mathbf{x}}_t^k - \eta \cdot \frac{\partial \mathcal{L}_{\text{obj}}}{\partial \hat{\mathbf{x}}_t^k}. \tag{8}$$

After $M$ steps optimization, we apply the standard denoising process to get the final result $\hat{\mathbf{x}}_0^{\text{edit}}$.

## 4 EXPERIMENTS

We validate our method with experiments and compare it with previous methods on indoor scene generation in accuracy, speed, and result quality. We further show various scene editing results to demonstrate the flexible control of our method.

### 4.1 EXPERIMENT SETUP

**Dataset:** We experiment on the 3D indoor scene dataset Structured3D (Zheng et al., 2020), which consists of 3,500 houses with 21,773 rooms designed by professional artists. Photo-realistic ren-

Table 1: Quantitative Comparison of panorama and mesh generation.

| Method | Panorama Metrics | | | | 2D Rendering Metrics | | 3D Mesh User Study | |
|---|---|---|---|---|---|---|---|---|
| | FID ↓ | CS ↑ | IS ↑ | Time/s ↓ | CS ↑ | IS ↑ | PQ↑ | 3DS ↑ |
| Text2Light (Chen et al., 2022) | 56.22 | 21.45 | **4.198** | 81.56 | - | - | 2.732 | 2.747 |
| MVDiffusion (Tang et al., 2023c) | 34.76 | **23.93** | 3.21 | 208.5 | - | - | 3.27 | 3.437 |
| Text2Room (Höllein et al., 2023) | - | - | - | ≥ 9,000 | 25.90 | 2.90 | 2.487 | 2.588 |
| Ours | **21.02** | 21.58 | 3.429 | **61.1** | 25.97 | 3.14 | **3.89** | **3.746** |

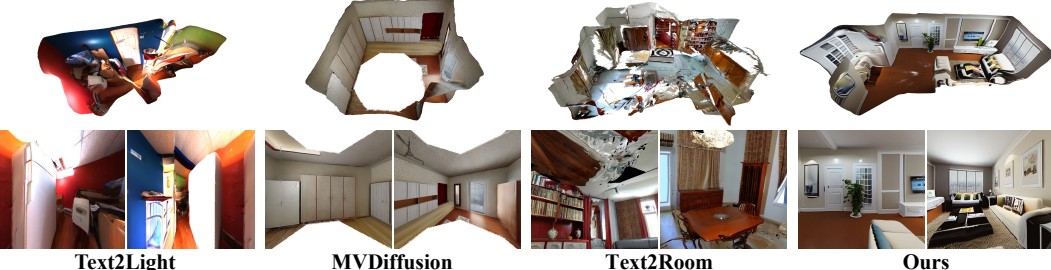

Figure 5: Qualitative comparison with previous works. For each method, we show a textured 3D mesh in the first row and a few rendered perspective images in the second row.

dered images, such as RGB panorama, semantic panorama, depth map, and normal map are provided in each room. Each room is also labeled with the room layout, represented by the intersection lines of walls, roofs, and floors. 3D bounding boxes of furniture items are provided but noisy. We parse these boxes for all the *'living rooms'* and *'bedrooms'*. Then, we follow (Wang et al., 2021) to generate text prompts describing the scene layout. Please refer to Appendix Sec.1.3 for more details about data preprocessing. The filtered dataset for training and evaluation includes 4,961 bedrooms and 3,039 living rooms. For each room type, we use $80\%$ of rooms for training and the remaining for testing.

**Metrics:** We choose Frechet Inception Distance (FID) (Heusel et al., 2017), CLIP Score (CS) (Radford et al., 2021), and Inception Score (IS) (Salimans et al., 2016) to measure the image quality of generated panoramas. We also compare the time cost to synthesize an RGB panorama of size $512 \times 1024$. To compare the quality of 3D room models, we follow Text2Room (Höllein et al., 2023) to render images of the 3D room model and measure the CLIP Score (CS) and Inception Score (IS). We further conduct a user study and ask 61 users to score Perceptual Quality (PQ) and 3D Structure Completeness (3DS) of the final room mesh on scores ranging from 1 to 5.

## 4.2 COMPARISON WITH PREVIOUS METHODS

**Quantitative Comparison** To evaluate our generated panoramic images, we follow MVDiffusion (Tang et al., 2023c) to crop perspective images from the 1,181 generated panoramas on the test split and evaluate the FID, CS, IS on the cropped multi-view images. These quantitative results are summarized on the left of Table 1, where our method achieves the best score in FID, which indicates that our method can better capture the room appearance because of its faithful recovery of the room layout. However, our score on CS is slightly lower than MVDiffusion, which seems to be insensitive to the number of objects as we illustrated in the supplementary file, and cannot reflect the quality of room layouts. The IS score depends on the semantic diversity of the cropped images as captured by an image classifier. It turns out Text2Light has the best IS score, since its generated indoor scenes often contain unexpected objects. More panoramic visualization can be found in Appendix Sec.1.5.

In terms of running time, our method takes the shortest time. Averagely speaking, our method takes only 61 seconds to generate a panorama, and another 20 seconds to generate the textured 3D mesh. In comparison, MVDiffusion takes 208 seconds, about 3.5 times longer, to synthesize a panorama. Text2Room needs at least 150 minutes to finish a textured 3D room generation.

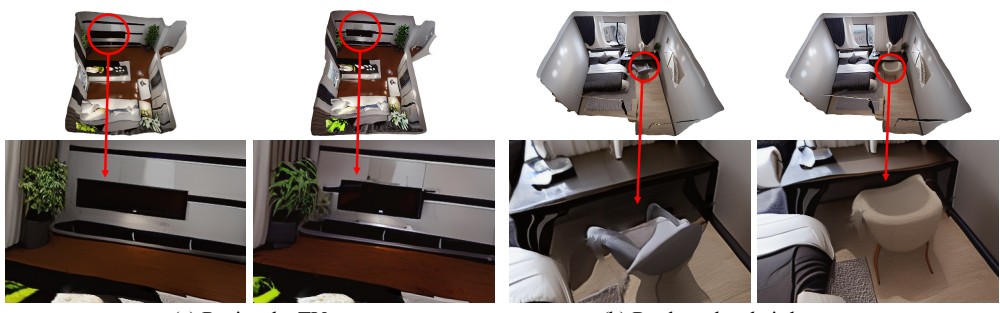

(a) Resize the TV  (b) Replace the chair by a new one

Figure 6: Room editing examples. (a) resize the TV, (b) replace the chair with a new one.

We then compare the 3D room models in terms of their rendered images. Because of the expensive running time of Text2Room (Höllein et al., 2023), we only test on 12 examples for this comparison. In this comparison, we further skip Text2light and MVDiffusion since we have compared them on panoramas. As the room layout is better captured with a large FOV, we render 60 perspective images of each scene with a 140° FOV and evaluate their CS and IS scores respectively. Please refer to Appendix Sec.1.4.1 for more details. The results of this comparison are shown in the middle of Table 1. Our method obtains better scores on both metrics than Text2Room.

We further evaluate the quality of the textured 3D mesh model by user studies. For those panorama generation methods, we utilize the depth estimation work (Shen et al., 2022) to reconstruct a textured 3D room mesh. More implementation details can be found in the Appendix Sec.1.3. The results of the user study are shown on the right of Table 1. Users prefer our method over others, for its clear room layout structure and furniture arrangement.

**Qualitative Comparison** Fig. 5 shows some results generated by different methods. The first row shows a textured 3D room model, and the second row shows some cropped perspective images from the panorama. As we can see, Text2Light (Chen et al., 2022) cannot generate a reasonable 3D indoor scene. It even fails to ensure the loop consistency of the generated panorama, which leads to distorted geometry and room model. Both MVDiffusion (Tang et al., 2023c) and Text2Room Höllein et al. (2023) can generate vivid local images as demonstrated by the perspective renderings in the second row. But they fail to capture the more global scale room layout. Similar effects can be seen from the Fig. 1 (a). These two methods often repeat a dominating object, e.g. a bed in the bedroom or fireplace in the living room, multiple times at different places and violate the room layout constraint. In comparison, our method does not suffer from these problems and generates high-quality results. More examples are provided in the supplementary file.

### 4.3 INTERACTIVE SCENE EDITING

We demonstrate the scene editing capability of our method in Fig. 6. In this case, we resize the TV and replace the chair in the generated results. Fig. 1 (b) shows two additional examples of replacing the TV and TV stand and moving the picture upwards. Our method can keep the visual appearance of the moved/resized objects unchanged after editing. More examples can be found in the supplementary file.

### 5 CONCLUSION AND DISCUSSION

We present **Ctrl-Room**, a flexible method to achieve editable and structurally plausible 3D indoor scene generation. It consists of two stages, the layout generation stage and the appearance generation stage. In the layout generation stage, we design a scene code to parameterize the scene layout and learn a text-conditioned diffusion model for text-driven layout generation. In the appearance generation stage, we fine-tune a ControlNet model to generate a vivid panorama image of the room with the guidance of the layout. Finally, a high-quality 3D room with a structurally plausible layout and realistic textures can be generated. We conduct extensive experiments to demonstrate that **Ctrl-Room** outperforms existing methods for 3D indoor scene generation both qualitatively and quantitatively, and supports interactive 3D scene editing.

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
