# CTRL-ROOM: CONTROLLABLE TEXT-TO-3D ROOM MESHES GENERATION WITH LAYOUT CONSTRAINTS (SUPPLEMENTARY MATERIALS)

## 1 APPENDIX

In the appendix, we first present more details about our scene code diffusion model in Sec. 1.1, then we provide our dataset pre-processing, text prompt generation, and implementation details in Sec. 1.2 and Sec. 1.3 respectively. Additional experiment results are illustrated in Sec. 1.4 and user studies in Sec. 1.6. Finally, we clarify the limitations of our method.

### 1.1 SCENE CODE DENOISING NETWORK

In the Layout Generation Stage, we use a holistic scene code to parametrize the indoor scene and design a diffusion model to learn its distribution. Specifically, given a 3D scene $\mathcal{S}$ with $N$ objects, we represent the scene layout as a holistic scene code $\mathbf{x_0} = \{\mathbf{o_i}\}_{i=1}^N$. We encode each object $o_i$ as a node with various attributes, i.e., center location $l_i \in \mathbb{R}^3$, size $s_i \in \mathbb{R}^3$, orientation $r_i \in \mathbb{R}$, class label $c_i \in \mathbb{R}^C$. Each node is characterized by the concatenation of these attributes as $\mathbf{o_i} = [c_i, l_i, s_i, r_i]$. As shown in Fig. 1, our scene code denoising network of the layout diffusion model is built upon IDDPM (Nichol & Dhariwal, 2021). The whole architecture of the layout diffusion model is similar to IDDPM, while we replace the upsample and downsample blocks with 1D-convolution network in the U-Net, and insert attention blocks after each residual block to capture both the global context among objects and the semantic context from the input text prompt. The input encoding head processes different encoding of the node attributes, e.g., semantic class labels, box centroid, and box orientation. After adding noise, the input encoding is fed into the U-Net to obtain a denoised scene code. During the forward phase, as in IDDPM, we iteratively perform the denoising process and generate a scene code from a partial scene textual description.

The goal of training the reverse diffusion process is to find optimal denoising network parameters that can generate natural and plausible scenes. The training objectives include the denoising objective $\mathcal{L}$ in Sec.3.1, which constrains the generated scene codes can approximate the underlying data distribution, and a regularization term $\mathcal{L}_{\text{physical}}$ to penalize the penetration among objects and walls.

$$\mathcal{L}_{\text{physical}} = \sum_{t=1}^{T} w_t * (\mathcal{L}_{\mathbf{w-o}} + \mathcal{L}_{\mathbf{o-o}})$$

$$\mathcal{L}_{\mathbf{w-o}} = \sum_{i=1}^{K_{\text{wall}}} \sum_{j=1}^{K_{\text{object}}} \sum_{p=1}^{8} Relu[-(a_i x_{jp} + b_i y_{jp} + c_i z_{jp} + d_i)]\mathbb{1}(\prod_{\mathbf{w}_i}(x_{jp}, y_{jp}, z_{jp})\text{in}\mathbf{w}_i) \quad (1)$$

$$\mathcal{L}_{\mathbf{o-o}} = \sum_{\mathbf{o}_i, \mathbf{o}_j}^{K_{\text{object}}} \mathbf{IoU}(\mathbf{o_i}, \mathbf{o_j})$$

where $\mathcal{L}_{\mathbf{w-o}}$ is physical violation loss between walls and objects. The $(a_i, b_i, c_i)$ is the normal vector of wall $\mathbf{w}_i$ that points to the room center. $\prod_{\mathbf{w}_i}$ means the operator projecting a point onto the plane that $\mathbf{w}_i$ defines. The plane equation of i-th wall is $a_i x + b_i y + c_i z + d = 0$ and $\mathbb{1}(\prod_{\mathbf{w}_i}(x_{jp}, y_{jp}, z_{jp})\text{in}\mathbf{w}_i)$ indicates whether the projection of bounding box vertice $(x_{jp}, y_{jp}, z_{jp})$

Table 1: Quantitative comparisons on the task of text-conditioned layout synthesis on the 3D-FRONT dining and living rooms. Note that for the Scene Classification Accuracy (SCA), the score closer to $50\%$ is better. The number for Diffuscene and ATISS is copied from DiffuScene paper.

| Method | Livingroom | | | Diningroom | | |
|---|---|---|---|---|---|---|
| | FID $\downarrow$ | KID $\uparrow$ | SCA | FID $\downarrow$ | KID $\uparrow$ | SCA |
| ATISS (Paschalidou et al., 2021) | 40.45 | 4.57 | 63.48 | 36.61 | 1.90 | 55.44 |
| DiffuScene (Tang et al., 2023a) | **35.27** | **0.64** | **54.69** | **32.87** | **0.57** | **51.67** |
| Ours | 36.0 | 1.4 | 56.42 | 34.78 | 1.3 | 54.37 |

of j-th object is in $\mathbf{w}_i$: if it is, return 1; otherwise, return 0. We skip some types of objects do intersect with the layout, such as windows and doors. Figure 2 depicts the physical loss among walls and objects vividly. $\mathcal{L}_{o-o}$ is the IoU summation of arbitrary two bounding boxes of objects. The hyperparamter $w_t$ is set to $\bar{\alpha}_{\mathbf{t}} * 0.1$

To verify the effectiveness of our first stage, we further conduct comparative experiments with these state-of-the-art layout synthesis approaches. Since 3D-FRONT (Fu et al., 2021) lacks walls, doors and windows annotation, we only use the furniture offered in 3D-FRONT to construct the scene code for each room. Table 1 shows a qualitative comparison with DiffuScene and ATISS, indicating that our method achieves results comparable to DiffuScene on the text-to-layout task. Although DiffuScene's results are slightly better than our qualitative results, this can be attributed to its use of an additional network to learn a shape code for each piece of furniture. This allows DiffuScene to retrieve a more accurate CAD model, improving its qualitative metrics.

Since most furniture in the room is associated with the walls, using different wall numbers in the text prompt should generate different rooms with different layouts. In Figure. 3, we provide additional visualizations of the generated scene layout in the format of 3D semantic bounding boxes. Our results demonstrate that users can control the number of walls or the layout of the room by changing the input text.

## 1.2 DATASET

**Structured3D dataset preprocessing** Structured3D consists of $3,500$ houses with $21,773$ rooms, where each room is designed by professional designers with rich 3D structure annotations, including the room planes, lines, junctions, and orientated bounding box of most furniture, and photo-realistic 2D renderings of the room. In our work, we use the 3D orientated bounding boxes of furniture, 2D RGB panorama, and 3D lines and planes of each room. While the original dataset lacks semantic class labels for each furniture bounding box. The dataset preprocessing aims to produce clean ground truth data for our layout generation module and appearance generation module.

- **Orientated Object Bounding Box Annotation.** As the original dataset lacks semantic label for each orientated object bounding box, we first unproject the RGB panorama and depth map into a point cloud of the room, then manually annotate the object semantic class and add more accurate object bounding boxes based on the noisy annotation of the original version. As shown in Fig. 4, by using labelCloud (Sager et al., 2022), three data annotators worked for 800 hours to annotate 5,064 bedrooms and 3064 livingrooms, getting nearly 50K and 69K accurate 3D bounding boxes across 25 object categories, respectively.

- **Scene Node Encoding.** We define our holistic scene code based on a unified encoding of walls and object bounding box. Each object $o_j$ is treated as a node with various attributes, i.e., center location $l_i \in \mathbb{R}^3$, size $s_i \in \mathbb{R}^3$, orientation $r_i \in \mathbb{R}$, class label $c_i \in \mathbb{R}^C$. The orientated bounding box is off-the-shelf, we extract the inner walls based on the line junctions and corners of the 3D room. Then we put the orientated object bounding boxes and walls into a compact scene code. Concretely, we define an additional 'empty' object and pad it into scenes to have a fixed number of object across scenes. Each object rotation

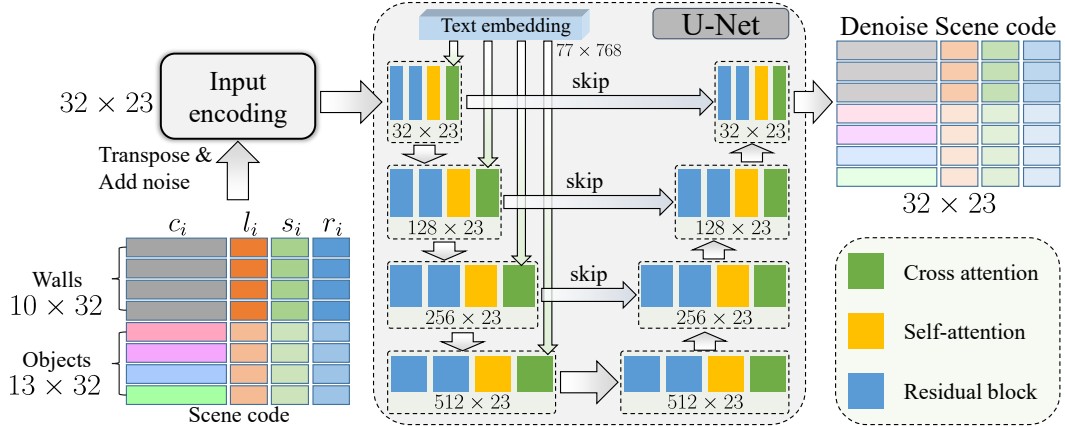

Figure 1: The detailed structure of the scene code denoising network. We here take the bedroom for example to demonstrate the scene code denoiser's dataflow. The scene code tensor $\mathbf{x_0} \in \mathbb{R}^{N \times D}$, where $N = 23, D = 32$.

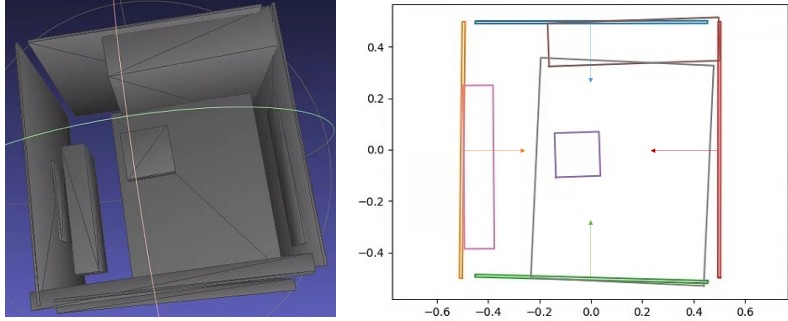

Figure 2: Objects-walls physical violation example. The physical violation loss is calculated only when the object intersection with wall. There is no violation loss when the object is completely inside or outside the walls.

angle is parametrized by a 2-d vector of cosine and sine values. Finally, each node is characterized by the concatenation of these attributes as $\mathbf{o}_i = [c_i, l_i, s_i, \cos r_i, \sin r_i]$.

- **data filtering.** We start by filtering out those problematic scenes such as rooms with wall number less than 4 or larger than 24. We also remove those scenes with too few or too many objects. The number of walls of valid bedrooms is between 4 and 10, and that of objects is between 3 and 13. As for living rooms, the minimum and maximum numbers of walls are set to 4 and 24, and that of objects are set to 3 and 21 respectively. Thus, the number of scene nodes is $N = 23$ in bedrooms and $N = 45$ in living rooms. After filtering, we get 4,961 bedrooms and 3,039 living rooms.

**Text Prompt Generation** We follow the SceneFormer Wang et al. (2021) to generate text prompts describing partial scene configurations. Each text prompt contains two to four sentences. The first sentence describes how many walls are in the room, then the second sentence describes two or three existing furniture in the room. The following sentences mainly describe the spatial relations among the furniture, please refer to SceneFormer Wang et al. (2021) and DiffuScene Tang et al. (2023a) for more detailed explanation of relation-describing sentences. In this way, we can get some relation-describing sentences to depict the partial scene. Finally, we randomly sampled zero to two relation-describing sentences to form the text prompt for 3D room generation.

## 1.3 IMPLEMENTATION DETAILS

**Training and inference details.**

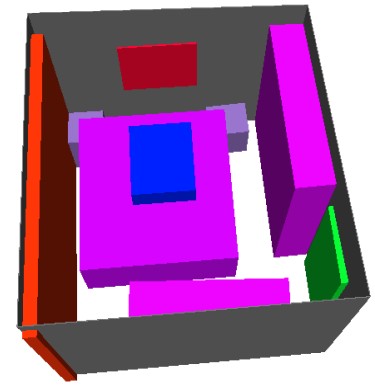

The bedroom has four walls.The room has two cabinets .

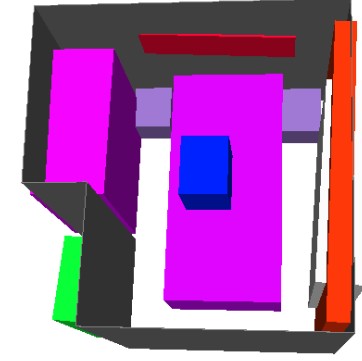

The bedroom has six walls.The room has a cabinet , a window and a bed .There is a lamp above the bed .There is a picture above the bed .

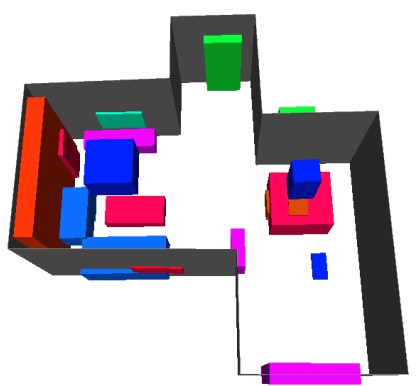

The living room has ten walls.The room has a cabinet , a window and a shelves .There is a lamp above the cabinet

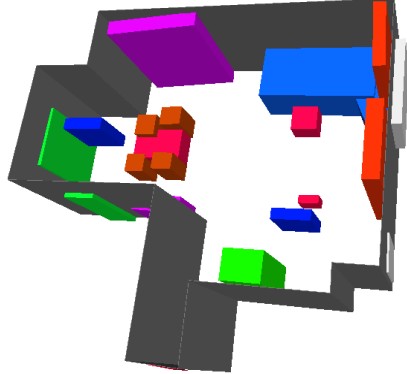

The living room has twelve walls.The room has a cabinet , a fridge and a window

Figure 3: Visualizations of the generated layout. The white box represents window, the burgundy boxe represents curtain, while the green box represents the door. Note that in the layout we generated, windows and curtains or doors intersect with the walls, which is reasonable since these objects are typically found on the walls. The room layout we generated is reasonable and consistent with the input text instructions. Users can modify the input text instructions to change the number and distribution of walls in the room.

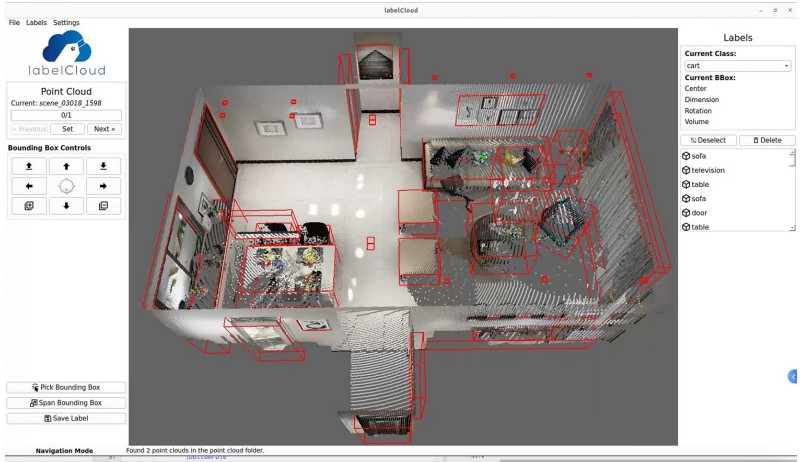

Figure 4: Example of object bounding box annotation.

- In the layout generation stage, We train the scene code diffusion model on our processed *'bedroom'* and *'livingroom'* data of Structured3D (Zheng et al., 2020) for $200,000$ steps. The frozen text encoder we adopted is the same as Stable Diffusion (**?**). The training is performed using the AdamW optimizer with a batch size of $128$ and a learning rate of $1e-4$, utilizing 2 A6000 GPUs. During the inference process, we utilize the DDIM (Song et al., 2020) sampler with a step size of $250$ to perform scene code denoising.

- In the appearance generation stage, we fine-tune the segmentation-conditional ControlNet model based on the pairwise semantic and RGB panorama of Structured3D. The fine-tuning process is implemented on two A6000 GPUs for $150$ epochs(about 3 days). In the inference phase, we generate high-fidelity and loop-consistent RGB panorama through DDIM sampler with $100$ steps, rotating both semantic layout panorama and the denoised image for $\gamma = 90°$ at each step.

- As for the mask-guided editing module, we utilize the fine-tuned Control-Seg model to inpaint the background content and optimize the latents of the edited panorama. In inpainting step, the weights used too fuse the unpainted area and unchanged area are set $\lambda_{\text{ori}} = 0.8, \lambda_{\text{new}} = 0.2$ . In the optimization step, the maximum iteration is $M = 50$, the learning rate $\eta$ for optimization is initialized to $0.1$ and then gradually decreases to $0.01$.

Table 2: ==The computational cost comparison on A6000. Since Text2Room does not offer a standardized neural network, we cannot measure its parameters.==

| Method | Inference Time/s | GPU Memory | Params |
|---|---|---|---|
| Text2Light | 81.56 | 5.46G | 630.66M |
| MVDiffusion | 208.5 | 8.74G | 1352.54M |
| Text2Room | $> 9,000$ | $> 16$G | - |
| Ours | 61.1 | 1.95G + 10.41G | 63.51M + 1220.62M |

### 1.3.1 BASELINE IMPLEMENTATIONS

We provide implementation details for baseline methods in the following:

- MVDiffusion (Tang et al., 2023b): To get a high-resolution photo realistic panorama, MVDiffusion employs $8$ branches of SD (Rombach et al., 2022) model and correspondence-aware attention mechanism to generate multi-view images simultaneously. We first fine-tune the pre-trained model of MVDiffusion on Structured3D for $10$ epochs(about 3 days). Since each generated subview image of MVDiffusion is at $512 \times 512$ resolution, the final panorama is pretty large. We resize the generated panorama of MVDiffusion from $4096 \times 2048$ to $1024 \times 512$. Then the $8$ subview perspective

images are extracted from the post-processed panorama using the same camera settings (FOV=90°,rotation=45°). The same operation is adopted on our generated panoramic images. Finally, we combine the panorama from MVDiffusion with our panoramic reconstruction module to create a 3D mesh.

- Text2Light (Chen et al., 2022): Text2Light creates HDR panoramic images from text using a multi-stage auto-regressive generative model. We choose Text2Light as one of the baseline for our panorama generation and 3D room mesh generation. We first generate RGB panoramas from the input text using Text2Light, then lift it into 3D mesh using the same panoramic reconstruction module as our method. When evaluate the panoramic image quality, we adopt the same processing as MVDiffusion to get multi-view perspective images of Text2Light.

- Text2Room (Höllein et al., 2023): Text2Room is the current state-of-the-art and off-the-shelf method for 3D room mesh generation. It utilizes 20 camera spots of a pre-defined trajectory to expand new areas as much as possible by generating 10 images at each spot. Here We use its final fused poison mesh for 3D mesh comparison. For a fair comparison of 2D renderings evaluation, we only use the renderings at the origin of the final mesh.

We also compare the model complexity between ours and the baselines, Tabel 2 shows our method achieves significantly faster runtime for generating a 3D room compared to existing methods. The difference in GPU memory footprint and parameter quantity is not significant. Despite having two diffusion models, our layout generation stage only needs to produce high-level room layouts and furniture arrangements, resulting in low computational costs and model complexity.

## 1.4 PANORAMA GENERATION COMPARISON

Fig. 5 presents additional results for panorama generation. Given a simple partial-scene text prompt, our approach obtains better RGB panorama than that of Text2Light (Chen et al., 2022) and MVDiffusion (Tang et al., 2023b), which demonstrates the effectiveness of our well-designed framework. While Text2Light suffers from the inconsistent loop and unexpected content of the generated panorama, MVDiffusion fails to recover a reasonable room layout from the text prompt.

## 1.5 ADDITIONAL QUALITATIVE RESULTS

We show additional qualitative comparison results between our method and baselines in Fig. 7. More demonstration of the visual quality of the generated geometry is shown in Fig. 8. We demonstrate more scene editing results of our method in Fig. 6.

## 1.6 USER STUDY

Follow Text2Room Höllein et al. (2023), we conduct a user study and ask $n = 61$ ordinary users to score the Perceptual Quality(PQ) and 3D Structure Completeness(3DS) of the generated room on a scale of $1 - 5$. Different from Text2room which only demonstrates the perspective renderings of the 3D room, we directly show users the generated mesh to get a global evaluation of the whole generated 3D room. We show an example of the presented interface of the user study in Fig. 9. In total, we presented 40 top-down views from 10 scenes and report averaged results for each method. Users favor our approach, which emphasizes the superiority of our more plausible geometry, along with the vivid texture.

## 1.7 LIMITATION

Although we have shown impressive 3D room generation results, there are still some limitations in our method. Firstly, we only support single-room generation, thus we cannot produce large-scale indoor scenes with multiple rooms. Secondly, the generated 3D room still contains incomplete structures in invisible area. Users might observe obviously stretched texture because of the occlusion and poor performance of the panoramic depth estimator. It will further affect the visual quality of scene editing results. For example, in Fig. 6(b), we find the table after movement is distorted and the tabletop becomes inclined, the problem is mainly caused by the inaccurate depth estimation and

**The bedroom has four walls. The room has a cabinet and a window.**

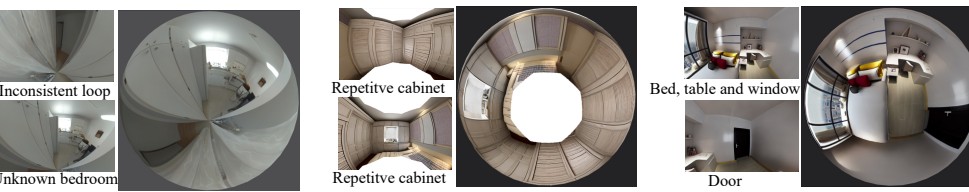

**The bedroom has eight walls. The room has two windows and a bed.**

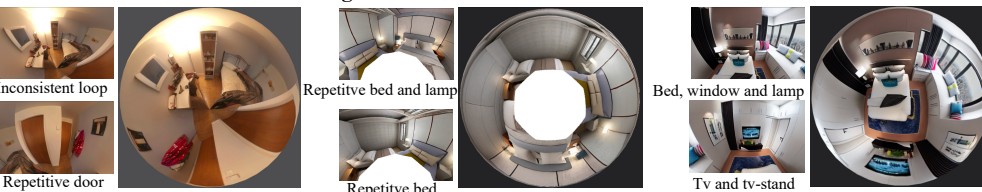

**The living room has ten walls. The room has a cabinet and a shelves .**

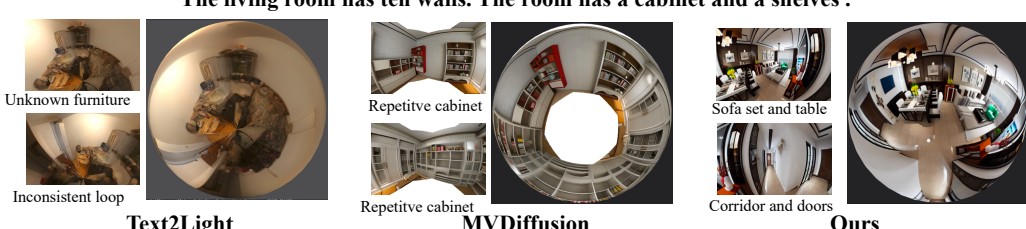

Figure 5: Qualitative comparison for panorama generation. Generated panorama is visualized in a panoramic image viewer to facilitate the user to check the global content of panorama. The left side of each column is two zoom-in views, and the right side is the fisheye view. Text2Light (Chen et al., 2022) exists serious inconsistent problem on the border of the generated panorama, it also shows a lot of unexpected stuff in the image. MVDiffusion (Tang et al., 2023b) fails to synthesize reasonable content for the target room type. In contrast, our method obtains layout plausible and vivid panorama from the given text prompt of partial scene.

the next mesh reconstruction and mesh texturing process. A promising direction is to learn a text-driven diffusion model to produce one or more RGB-D panorama images under the scene layout constraints. We leave these mentioned limitations as our future efforts.

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

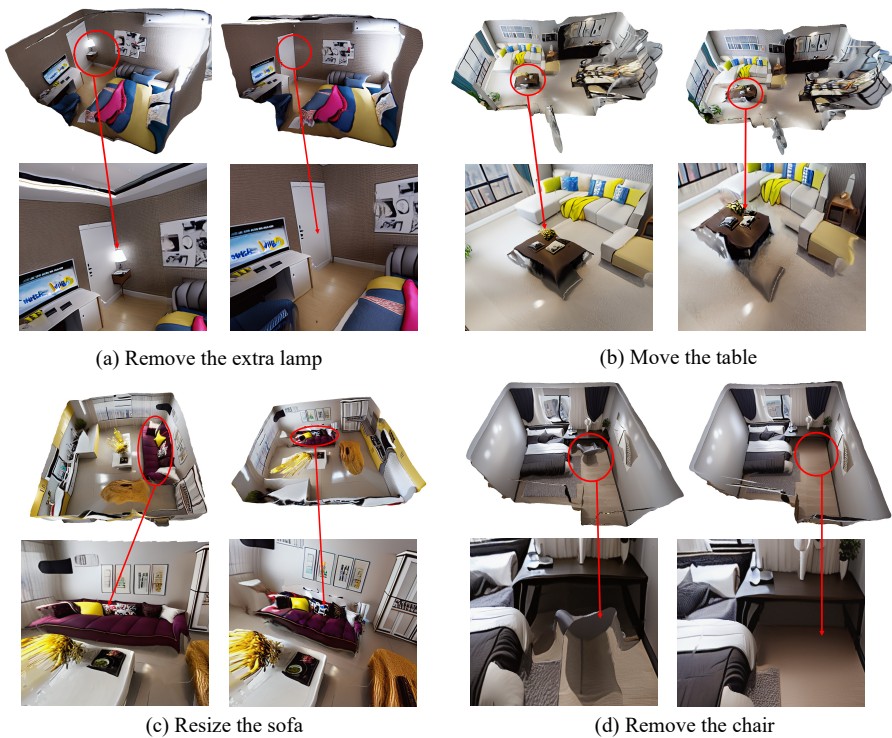

(a) Remove the extra lamp

(b) Move the table

(c) Resize the sofa

(d) Remove the chair

Figure 6: Additional scene editing results. In each sub-figure, the left part is the original 3D room, the right part shows the final mesh after users' interactive editing.

Robin Rombach, Andreas Blattmann, Dominik Lorenz, Patrick Esser, and Björn Ommer. High-resolution image synthesis with latent diffusion models. In *Proc. CVPR*, pp. 10684–10695, 2022.

Christoph Sager, Patrick Zschech, and Niklas Kuhl. labelCloud: A lightweight labeling tool for domain-agnostic 3d object detection in point clouds. *Computer-Aided Design and Applications*, 19(6):1191–1206, mar 2022. doi: 10.14733/cadaps.2022.1191-1206. URL `http://cad-journal.net/files/vol_19/CAD_19(6)_2022_1191-1206.pdf`.

Jiaming Song, Chenlin Meng, and Stefano Ermon. Denoising diffusion implicit models. *arXiv preprint arXiv:2010.02502*, 2020.

Jiapeng Tang, Yinyu Nie, Lev Markhasin, Angela Dai, Justus Thies, and Matthias Nießner. Diffuscene: Scene graph denoising diffusion probabilistic model for generative indoor scene synthesis. *arXiv preprint arXiv:2303.14207*, 2023a.

Shitao Tang, Fuyang Zhang, Jiacheng Chen, Peng Wang, and Yasutaka Furukawa. Mvdiffusion: Enabling holistic multi-view image generation with correspondence-aware diffusion. *arXiv preprint arXiv:2307.01097*, 2023b.

Xinpeng Wang, Chandan Yeshwanth, and Matthias Nießner. Sceneformer: Indoor scene generation with transformers. In *2021 International Conference on 3D Vision (3DV)*, pp. 106–115. IEEE, 2021.

Jia Zheng, Junfei Zhang, Jing Li, Rui Tang, Shenghua Gao, and Zihan Zhou. Structured3d: A large photo-realistic dataset for structured 3d modeling. In *Proc. ECCV*, pp. 519–535. Springer, 2020.

**The bedroom has six walls. The room has a cabinet and a window.**

| Text2Light | MVDiffusion | Text2Room | Ours |

**The bedroom has four walls. The room has a window and a picture.**

| Text2Light | MVDiffusion | Text2Room | Ours |

**The bedroom has four walls. The room has a cabinet and a window.**

| Text2Light | MVDiffusion | Text2Room | Ours |

**The living room has ten walls. The room has a picture and a window.**

| Text2Light | MVDiffusion | Text2Room | Ours |

Figure 7: Additional qualitative comparison with previous works.

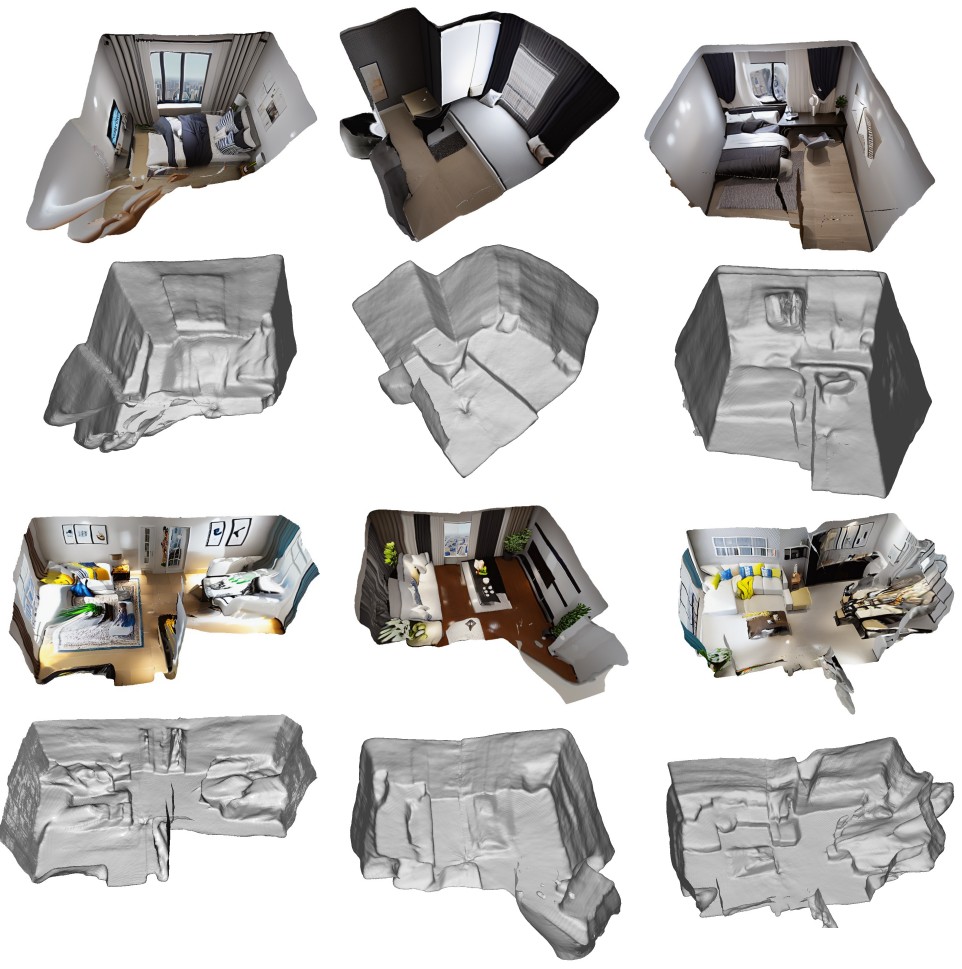

Figure 8: Visualization of generated geometry of the 3D scenes. We show color renderings from generated scenes at the first and third rows, and the corresponding shaded geometry renderings at the second and fourth rows.

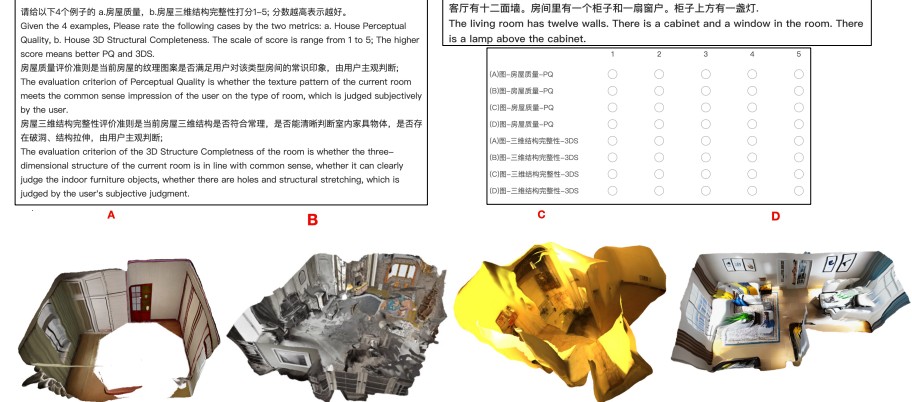

Figure 9: User study interface. We provide users with multiple top-down images from different methods and ask users to rate the given 3D meshes on a scale from 1 to 5, according to the criteria of Perceptual Quality and 3D Structure Completeness.

**The living room has twelve walls. The room has a cabinet and a fridge**

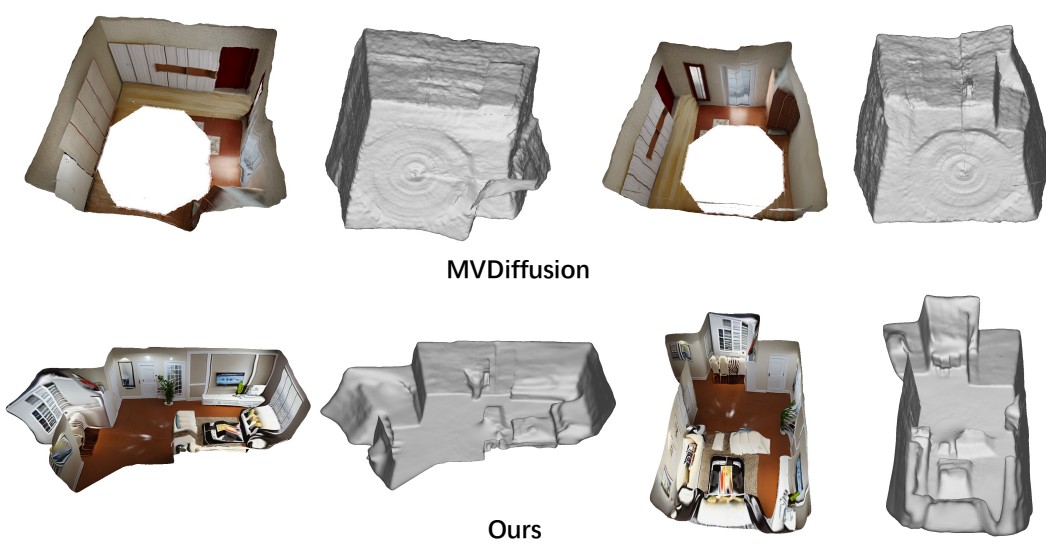

MVDiffusion

Ours

Figure 10: Comparison of geometry quality between our result and MVDiffusion's. The first and third columns are color renderings from different views, and the second and last columns are the shaded geometry rendering. It is obvious that MVDiffusion results in an entire room filled with cabinets, whereas our approach is capable of generating a realistic room layout with various pieces of furniture, such as a TV, a table, and a sofa, all arranged sensibly in accordance with the input textual instructions. Furthermore, while MVDiffusion fails to generate walls with the correct number consistent with the input text, we can recover the required walls approximately. This highlights the effectiveness of our approach in generating more accurate room layouts.