# OpenReview forum: "Ctrl-Room: Controllable Text-to-3D Room Meshes Generation with Layout Constraints"
_ICLR.cc/2024/Conference — ICLR 2024 Conference Withdrawn Submission_

### Official Review · Reviewer_Bzvn · 2023-10-27

**Soundness:** 3 good
**Presentation:** 3 good
**Contribution:** 2 fair
**Rating:** 6
**Confidence:** 4

**Summary:**

This paper proposes a controllable method for room-scale text-to-3D meshes: it first utilises scene layout generation diffusion model to generate 3D scene layouts from text prompts, and then generates panorama given rendered semantic layout as conditions. With this design, it could control the object arrangements  of generated 3D meshes, and it is easy to edit the generated 3D meshes ad hoc.

**Strengths:**

1. Adding controls into scene-scale 3D generation is a promising direction, and this paper gives the first several attempts to solve it. The whole pipeline is reasonable  and sound.
2. The editing experiment is impressive and interesting.

**Weaknesses:**

1. The variety of generation. Since both scene generator and controlnet are trained on Structure3D dataset, which contains only "living room" and "bedroom", I am concerned the variety of scenes this method could generate. I am also curious about how does it work on some other rooms, like "bathroom", "kitchen" and others?
2. Text prompts are strange.  The text prompts shown in this paper are not very natural. For example, " the living room has eight walls,. The room has a picture, a shelves and a cabinet".  These prompts are very different from prompts people typically use to describe the rooms. I am curious about how it works with  more natural and rand descriptions? Like "a living room with a cozy, yellow coach and red curtain" and others.
3. Related to point 2,  the given prompts specifies the number of walls in the room. Is it necessary to specify the wall numbers to get good results? What is the quality of scene generation? Are the number of walls generated the same as number in text prompts? Counting the number of wall instances in generated layout and comparing to numbers in text prompts might be an  interesting metric to look on.
4. Not a weakness but some citations are missing:
[1] CC3D: Layout-Conditioned Generation of Compositional 3D Scenes
[2] RoomDreamer: Text-Driven 3D Indoor Scene Synthesis with Coherent Geometry and Texture

**Questions:**

1. To measure the variety of generations, it might be helpful to also report the FID/KID on scene layout generations, as DiffuScene, which could be a reflection about scene layout variety.
2. More results on diverse prompts would be more convincing, including (1): prompt style: removing xx walls, adding more adjective and using more natural language; (2) trying contents beyond "living room", "bedroom".

---

> ### Author Response · Authors · 2023-11-20
> **Reply to Bzvn**
>
> Thanks for your comments. We address the weakness and questions below:
>
> * **Variety of text prompt and generated room type**: Many thanks for your valuable suggestions. Since our layout diffusion model is only trained on the annotated livingroom and bedroom with the fixed style text prompt, the layout distribution varies from different kinds of room, our framework could not generalize to the common prompt style and other types of room directly. We leave the generation of layouts with free-style text prompts for future work. For the original noisy Structured3D dataset, we have worked with three professional annotators for 800 hours to annotate 5,064 bedrooms and 3,064 living rooms, getting nearly 50K and 69K accurate 3D bounding boxes across 25 object categories, respectively. We plan to release our annotated living room and bedroom datasets soon, and are also working on expanding to other types of rooms. Hopefully, these datasets would facilitate more research in this direction.
>
> * **Text prompts are strange**: Labeling text descriptions for high-quality complex scenes with a large number of objects is difficult, so we use the same rule as SceneFormer~\citep{wang2021sceneformer} to produce the text prompt for each room. The text prompt only describes a partial scene configuration, including several walls, objects, and their spatial relationships. Please refer to Section 1.2 in the appendix for more information on how to produce the text prompt. We acknowledge that extending our method to open-vocabulary textual input would be an interesting direction for future research.
>
> * **Counting wall numbers in generated layout**:  Since most furniture in the room is associated with the walls, using different wall numbers in the text prompt should generate rooms with different layouts. In Figure.3 of the revised Appendix, we provide additional visualizations of the generated scene layout in the format of 3D semantic bounding boxes. Our results demonstrate that users can control the number of walls or the layout of the room by changing the input text.
>
> * **Missing Citations**: Thanks for your reminder.  We have added these works to our main paper.
>
> * **Measure the variety of layout generations.**: In the initial submission, we did not include a quantitative comparison for our first stage method, since our method considers both walls, doors, windows, and other furniture, while DiffuScene only considers furniture. Furthermore, DiffuScene has not released the text-conditional training code, making it difficult to compare our method with it on the Structured3D dataset. In hindsight, we agree with the reviewer that this comparison is relevant and include such a comparison on the 3D-FRONT dataset in the Appendix Sec. 1.1. Table.1 shows a quantitative comparison with DiffuScene on 3D-FRONT, indicating that our method achieves results comparable to DiffuScene on the text-to-layout task. DiffuScene's results are slightly better because it uses an additional network to learn a shape code for each furniture, facilitating the inference process to retrieve a more accurate CAD model for each furniture and improving its quantitative metrics.

---

> > ### Author Response · Authors · 2023-11-23
> > **Additional questions?**
> >
> > We hope our response addresses your questions; if so, we would appreciate it if you would reconsider your score accordingly. As the deadline is approaching, please let us know soon if you have additional questions or concerns.

---

> > ### Comment · Reviewer_Bzvn · 2023-12-04
> > **Reply to authors' feedback**
> >
> > Thanks for the valuable feedback and they help a lot. After reading all the comments from other reviewers, I am positive about this paper. Please update the paper according to other reviewers' suggestions if it is accepted. Thanks

---

### Official Review · Reviewer_DxGi · 2023-10-31

**Soundness:** 2 fair
**Presentation:** 2 fair
**Contribution:** 2 fair
**Rating:** 5
**Confidence:** 4

**Summary:**

The paper proposes "Ctrl-Room," a method for text-driven 3D indoor scene generation. The authors present a two-stage approach where the room layout is first generated from text input and subsequently used to guide the appearance generation. The primary insight is to separate the modeling of layouts and appearance, which facilitates manual editing of layouts. Experiments are conducted primarily on the Structured3D dataset, demonstrating the method's potential in generating detailed, editable, and consistent 3D rooms from natural language prompts.

**Strengths:**

(+)  The intuitive approach of generating the layout first and then the appearance allows for fine-grained control over the generated scenes, enabling easy human intervention in editing the layout.

(+) The method, to some extent, avoids problems faced by existing methods, such as generating multiple beds in the same room, ensuring more realistic scene generation.

(+) The paper introduces the concept of loop consistency sampling, ensuring that the generated panoramic images maintain their cyclic consistency, especially at the edges.

**Weaknesses:**

(-) The method heavily relies on 3D bounding box annotations. Given the scarcity of datasets with such 3D annotations, the generalization capability of the text-to-layout process is limited. The experiments are restricted to generating only living rooms and bedrooms, without exploring the generation of other room types.

(-) As mentioned in the appendix, the current approach can only generate textures for a single panoramic image. It doesn't support multi-viewpoint generation, which limits the visual quality when the viewpoint is freely moved.

(-) The results for the layout generation stage lack sufficient evaluation and comparison, making it challenging to assess the effectiveness of the first stage in isolation.

**Questions:**

- In the first stage, all objects are represented using cuboid bounding boxes. How does this representation impact objects with a more "concave" shape like tables or desks? In these cases, the projected semantic mask would differ from the actual silhouette, how might this affect the ControlNet's performance in the second stage?
- In section 3.3, under "Optimization Step," the designed optimization target seems to only allow for object movement and scaling. Does the method support other types of edits?
- The paper doesn't demonstrate the visual quality of the generated geometry/mesh. What is the quality of the generated geometry, both in terms of accuracy and visual appeal?

---

> ### Author Response · Authors · 2023-11-21
> **Reply to DxGi**
>
> Thanks for your comments. We address the weakness and questions below:
>
> * **Relying on 3D annotation**: We appreciate the constructive feedback provided by the reviewer. As mentioned in the third response to reviewer eF7g, we fully concur that one of the key to ensuring a reasonable distribution of generated rooms lies in the layout generation component. Generating a sensible room layout without the introduction of annotated data as priors has been a challenge, as existing methods tend to suffer from the ``Penrose Triangle" problem (i.e. the multi-view consistency or the layout consistency problem), resulting in repetitive textures or objects in the generated rooms.
> To address this issue, we propose a framework where we first generate the layout and then generate 3D room content based on the generated layout. In order to ensure that the generated room layouts are both realistic and reasonable, we invested considerable effort in annotating a substantial amount of data (Three data annotators worked for 800 hours to annotate 5,064 bedrooms and 3,064 living rooms, getting nearly 50K and 69K accurate 3D bounding boxes across 25 object categories, respectively.). While our current experiments have primarily focused on living rooms and bedrooms, as pointed out by the reviewer,  we plan to expand the datasets to various room types in the future. Moreover, we are committed to making our code and all associated datasets publicly available. We hope that our contributions will prove beneficial for future research endeavors and further advancements in this field.
>
> * **Does not support random view rendering**: We greatly appreciate the valuable feedback from the reviewers. As we have mentioned in our main paper and supplementary materials, our current method is designed to generate panoramic views and does not support the generation of multiple perspectives. In the context of text-based 3D room generation, we believe that the primary challenge lies in overcoming the Penrose Triangle problem to generate rooms with realistic and reasonable layouts. Therefore, our paper introduces a two-stage framework to separate the generation of layout and appearance. Based on the generated results, our approach demonstrates the ability to produce more realistic and plausible outcomes compared to existing methods.
> We fully acknowledge and appreciate your suggestion. However, we believe that this is not the primary focus of the current paper. We plan to explore the generation of realistic freeform 3D rooms in future work, building upon the foundation laid out in this paper.
>
> * **Varify the effectiveness of the first stage**: In the initial submission, we did not include a quantitative comparison to those layout synthesis works(e.g. ATISS\citep{paschalidou2021atiss}, DiffuScene\citep{tang2023diffuscene}) for our first stage method, since our method considers both walls, doors, windows, and other furniture, while DiffuScene only considers furniture and cannot represent a closed indoor scene. Furthermore, DiffuScene has not released the text-conditional training code, making it difficult to compare our method with it on the Structured3D dataset. In hindsight, we agree with the reviewer that this comparison is relevant and include such a comparison on the 3D-FRONT dataset in the Appendix Sec. 1.1. Table.1 shows a quantitative comparison with DiffuScene on 3D-FRONT, indicating that our method achieves results comparable to DiffuScene on the text-to-layout task. Although DiffuScene's results are slightly better than our qualitative results, this can be attributed to its use of an additional network to learn a shape code for each furniture. This allows DiffuScene to retrieve a more accurate CAD model, improving its qualitative metrics.
> We have also provided additional visualization results for our first stage, which can be found in Figure 3 in supplymentary. From Figure 3, it can be observed that the layouts we generated are reasonable and almost conform to the distribution of real-world scenes.

---

> > ### Author Response · Authors · 2023-11-21
> > **Reply to DxGi**
> >
> > * **How the cubic bounding box impact first and second stage.**: We appreciate the reviewers' thorough understanding and valuable insights. Indeed, in the first stage of our framework, we primarily focus on generating the layout of the room, which includes aspects such as object categories, sizes, positions, and orientations. We do not explicitly consider the specific shapes of objects in this stage. Generating detailed object shapes in the first stage is indeed a challenging task and would likely require a substantial amount of finely segmented annotated data, which is significantly more difficult to obtain compared to bounding box (bbox) annotations. Therefore, in our approach, we generate high-level representations, the orientated object bounding boxes, in the first stage and use them to guide the second-stage panoramic image generation. As pointed out by the reviewer, semantic maps obtained from bboxes may differ from the actual silhouette generated. However, in our experiments, we have found that this discrepancy does not negatively impact our generation results (refer to Table .1 in the main paper and Figure.5 in the supplementary). As demonstrated in our main text and supplementary materials, using generated bounding boxes to obtain coarse semantic maps, while lacking fine-grained segmentation edges, still allows our trained network to generate detailed panoramic images with clear object boundaries. Furthermore, using this kind of semantic segmentation image ensures that each object in the panoramic image generated in the second stage has more diverse appearances, rather than being limited by the object silhouette as in the case of using fine-grained object segmentation images.
> >
> > * **Does the optimization step support other type edits**: Thank you for raising this concern. We will further clarify these ambiguous issues in the main paper. We would like to clarify that the optimization step is only used to maintain the consistent texture of objects, which is not considered in remove or replacement edits.  For remove and replace editing, we only use the inpainting step to ensure that the inpainted area has a seamless background in the original generated panorama.
> >
> > * **More visualizations of generated geometry**: In the appendix Figure.8, We have added more visualization results for the geometry of our generated 3D room. Since there is no ground truth in text-to-room task, we cannot report its’ accuracy. But our user study Table.1 in the main paper, proves that users prefer our method over others, for its clear room layout structure, furniture arrangement, and impressive appearance.

---

> > > ### Comment · Reviewer_DxGi · 2023-11-22
> > >
> > > Thank you for your detailed rebuttal addressing the concerns raised in my initial review. Your efforts in enhancing the manuscript and clarifying key aspects of your method are appreciated.
> > >
> > > However, after careful consideration, I am inclined to maintain my original score. This decision is primarily due to two factors: the method's limited generalization, as it currently applies mainly to living rooms and bedrooms, and its restriction to generating single panoramic images.

---

> ### Author Response · Authors · 2023-11-23
> **Reply to DxGi**
>
> Thank you for your comment.
> We want to further response on the two points raised by the reviewer.
> 1. currently applies only to living rooms and bedrooms:  As we stated earlier, we labeled 5K bedrooms and 3K living rooms by hiring 3 annotators working for 800 hours. This labeling process is expensive and time-consuming. Thus, we have only labeled living rooms and bedrooms. This dataset is prepared to verify the effectiveness of our proposed algorithm. It is not our focus to build a larger dataset with all kinds of rooms. Our dataset can be easily expanded and our method can be directly generalized to cover other types of rooms.
>
> 2. restriction to generating panorama images: We want to highlight that generating high quality panorama images for indoor scenes is not a simple task. As we demonstrated in the paper and the rebuttal, the state-of-the-art method MVDiffusion still suffers from the 'Penrose Triangle' problem, which violates the room layout constraints and generates multiple beds in a bedroom and multiple cabinets in a living room. Our key contribution is to introducing a stage for layout synthesis before the panorama (or 3D scene) generation to ensure high quality results.

---

### Official Review · Reviewer_eF7g · 2023-11-01

**Soundness:** 3 good
**Presentation:** 3 good
**Contribution:** 3 good
**Rating:** 6
**Confidence:** 5

**Summary:**

This paper proposes a method to generate 3D rooms from text. The proposed method separates the generation of layout and appearance into two stages. In the first stage, a text-condition diffusion is trained to obtain the scene code parameterization. In the second stage, a fine-tuned ControlNet is utilized to generate a room panoramic image. The experiments demonstrate they can generate view-consistent and editable 3D rooms from text.

**Strengths:**

1. The proposed method can locally control the 3D room generation, which can generate plausible indoor scenes.

2. The proposed method separates the layout generation and appearance generation.

3. The proposed method can achieve 3D indoor scene editing.

**Weaknesses:**

1. There are two diffusion models, which will lead to both large computation costs and GPU memory costs.

2. Although the rendering views seem better, the geometry seems worse than the MVDifffusion based on Figure 5.

**Questions:**

1. Figure 5 shows that the proposed method generates obviously worse results on the left walls compared to MVDiffusion. How did the user study show better results in Table 1?

2. The global consistency seems guided by the generated layout. How does the proposed method guarantee plausible consistency for the layout generation? For example, how does the proposed method guarantee two objects are not cross-overlapped?

---

> ### Author Response · Authors · 2023-11-20
> **Reply to eF7g**
>
> Thanks for your comments. We address the weakness and questions below:
>
> * **Computational Cost**: We appreciate the reviewer's suggestion, and as a response, we have added a table to compare our method with existing approaches in terms of runtime, parameter count, and GPU memory usage. As shown in Table.2 in the Appendix, our method achieves significantly faster runtime for generating a 3D room compared to existing methods. The GPU memory consumption and model parameters of ours are similar to MVDiffusion's. Despite having two diffusion models, our layout generation stage only needs to produce high-level room layouts and furniture arrangements, resulting in low computational costs and model complexity.
>
> * **Geometry Comparison in Figure5**: In Figure 5 of the main paper, certain views may appear geometrically incorrect due to depth estimation errors and heavy occlusion in the generated panorama. In the newly added Figure.10 in Appendix, we present an alternative view using mesh rendering and shaded geometry to compare with the results of MVDiffusion. From the results, it is evident that MVDiffusion results in an entire room filled with cabinets, whereas our approach is capable of generating a realistic room layout with various pieces of furniture, such as a TV, a table, and a sofa, all arranged sensibly in accordance with the input textual instructions. This highlights the primary issue this paper aims to address: achieving the generation of genuinely realistic and reasonable 3D rooms through a two-stage process of layout generation followed by appearance generation based on the layout. Furthermore, refer to Reviewer Bzvn.3, while MVDiffusion fails to generate walls with the correct number, we can recover the required walls approximately. This highlights the effectiveness of our approach in generating more accurate room layouts.
>
> * **Layout Consistency**: Thank you for your valuable feedback. It is indeed important to ensure consistency in the layout generation, since the final result is guided by this layout. To achieve this, we have adopted a diffusion-based generation method, which has been proven effective in capturing distributions and generating reliable layouts in other domains, such as 2D image generation. Please note that our layout diffusion model is trained on the scene code space, which consists of 3D furniture bounding boxes.
> To train this diffusion module for layout generation, we invested substantial efforts in annotating the 3D bounding boxes based on Structured3D dataset (three data annotators worked for 800 hours to annotate 5,064 bedrooms and 3,064 living rooms, getting nearly 50K and 69K accurate 3D bounding boxes across 25 object categories, respectively.).
> By training our layout diffusion model on such a finely annotated dataset, we are able to generate room layouts that closely adhere to real-world distribution patterns, as supported by our experimental results in Tabel.1 in the Appendix. Our method achieves a comparable result to the state-of-the-art method on text-to-layout generation task.
> On the other hand, during the training of our layout generation network, we have implemented a physical regularization loss to ensure that different objects do not cross overlap. This regularization penalty helps to maintain spatial relationships among objects, preventing undesired overlaps in adjacent areas of the rooms.
> We have provided more in-depth details in our revised version, please refer to Sec.1.1 in supplementary for a comprehensive understanding of our physical regularization loss.

---

> > ### Author Response · Authors · 2023-11-23
> > **Additional questions?**
> >
> > We hope our response addresses your questions; if so, we would appreciate it if you would reconsider your score accordingly. Please let us know soon if you have additional questions or concerns.

---

### Official Review · Reviewer_83GV · 2023-11-01

**Soundness:** 2 fair
**Presentation:** 3 good
**Contribution:** 2 fair
**Rating:** 5
**Confidence:** 5

**Summary:**

This paper proposes a two-stage generative modeling method for indoor scenes. In the first stage, the scene code is generated according to the text prompt, and then the scene code is converted into 3D geometry with bounding boxes with semantic class labels. The rendering of semantic 3D indoor scenes is used to generate panorama images as the texture of the indoor scene. Overall, I feel that this paper is a system work that applies diffusion models and control net to the task of 3D indoor scene generation.

**Strengths:**

1. The experimental results are impressive.
2. The two-stage design ease the editing of the generated 3D indoor scenes.

**Weaknesses:**

Lack of technical novelty.  Although two-stage design has its own advantages, it is more like a design strategy than a solid technical contribution, since such design is widely used in 3D content generation. For instance, visual object networks first generate geometry through a shape network and then generate rendering results through a texture network.

**Questions:**

I can still find some artifacts in the generated 3D rooms, such as the distortion of textures. Scene code is an approximation to the real geometry, is it possible to add another stage to convert it into freeform 3D surfaces that are ubiquitous in indoor scenes?

---

> ### Author Response · Authors · 2023-11-20
> **Reply to 83GV**
>
> Thanks for your comments. We address the weakness and questions below:
>
> * **Lack of novelty** : As we described in the second paragraph of the Introduction, the focus of this paper is to solve the multi-view consistency or layout consistency in previous text-to-room methods, which we referred to as the **Penrose Triangle Problem**. As shown in Fig 1 (a), previous methods often generate repetitive objects (e.g. multiple beds or multiple cabinets) in their results. Our method can effectively solve this problem and further allows intuitive editing of the generated model as shown in Fig 1 (b). Our method is simple and effective. No previous text-to-room methods have adopted a similar approach. While some text-to-objects generation methods also use a two-stage design, our method is rather different from these methods. Our shape generation stage involves diffusion in an abstract scene code space, and the appearance stage employs a ControlNet with the guidance of semantic segmentation maps. Quantitative and qualitative comparisons demonstrate the superiority of our method over previous text-to-room generation approaches.
>
> * **Artifacts and free-form object surface**: We greatly appreciate the detailed and insightful feedback from the reviewer. As discussed in limitation section (Appendix Sec 1.7), due to performance of the panoramic depth estimation network we are using, there may be some artifacts present in the 3D results we generated. Again, the focus of this work is to solve the Penrose Triangle Problem, e.g. the multi-view consistency or layout consistency. The 3D result can be improved with a more advanced depth estimation module, or as the reviewer suggested adding "another stage to convert it into freeform 3D surfaces".

---

> > ### Author Response · Authors · 2023-11-23
> > **Additional questions?**
> >
> > We hope our response addresses your questions; if so, we would appreciate it if you would reconsider your score accordingly. As the deadline is approaching, please let us know soon if you have additional questions or concerns.

---

### Author Response · Authors · 2023-11-20
**General Comment**

We thank all reviewers for their thoughtful comments.

We have uploaded a revised version of the main paper and supplementary for clarity and for several of the suggested changes. Rewritten regions are highlighted in yellow. We have added the missing citations in the related work as suggested by Bzvn. In the revised supplementary file, we added Figure 2 and10 to explain eF7g 's question, and added three new figures for qualitative comparisons (Figures 3 and 8), as suggested by DxGi, Bzvn.
 We hope that these substantial efforts to clarify and improve the paper will more clearly explain our contributions and prevent any misunderstandings.

We respond to each reviewer’s specific concerns in our individual responses.